# Long-Term Relationship between Psychological Distress and Continuous Sedentary Behavior in Healthy Older Adults: A Three Panel Study

**DOI:** 10.3390/medicina55090555

**Published:** 2019-08-31

**Authors:** Yutaka Owari, Nobuyuki Miyatake

**Affiliations:** 1Shikoku Medical College, Utazu, Kagawa 769-0205, Japan; 2Department of Hygiene, Faculty of Medicine, Kagawa University, Miki, Kagawa 761-0793, Japan

**Keywords:** long-term relationship, healthy elderly people, psychological distress, continuous sedentary behaviors, structural equation modelling (SEM)

## Abstract

*Background and objectives*: Psychological distress (PD) is associated with continuous sedentary behaviors (CSB; based on the ratio of 1.5 metabolic equivalents (METs) sessions or more continuing for over 30 min) in older adults, but the long-term relation is not sufficiently clarified. This study aims to clarify the long-term relationship between PD and the rate of CSB. *Materials and Methods*: In this secondary analysis, a sample population of 72 healthy elderly people aged 65 years or older participated in a health club of college A from 2016 to 2018. We conducted structural equation modeling (SEM) using the cross-lagged and synchronous effects models. We adopted the following as proxy variables: CSB and PD (based on the Kessler 6 scale (K6) scores). *Results*: “2016 K6” to“2017 CSB” (standardization factor (β = 0.141, *p* = 0.025), “2017 K6” to “2018 CSB” (β = 0.187, *p* < 0.001) and “2016 CSB” to “2018 CSB” (β = 0.188, *p* < 0.001) were all statistically significant using the cross-lagged effects models. Fit indices were adjusted goodness of fit index (AGFI) = 0.961, comparative fit index (CFI) = 1.000, and root mean square error of approximation (RMSEA) = 0.000. *Conclusion*: The results suggest that PD may affect the ratio of CSB one year later, and CSB may affect the ratio of PD two year later.

## 1. Introduction

Sedentary behavior has become incorporated into many aspects of both workplace and family life in modern societies. In particularly, the sedentary behavior of Japanese is the highest in the world [1,2]. Moreover, sedentary behavior increases with age [3] and is a risk factor for various adverse health outcomes [4,5]. In our study, sedentary behavior is defined not as “physical inactivity” (the definition of the World Health Organization [6]), but as “any waking behavior characterized by an energy expenditure ≤1.5 metabolic equivalents (METs) while in a sitting or reclining posture” [7].

Sitting too much has been associated with low mental health in elderly people [8,9,10]. Recent research has shown that oversitting is a health risk factor, even with moderate physical activity [4]. Japan has one of the largest elderly populations in the world. According to the Cabinet Office Government of the Japanese government, the aging rate of Japan (the percentage of the population over 65 years old) is 27.7% [11]. Therefore, maintaining the health of the elderly is important, and solving this problem would also be valid to other countries where the aging of society is progressing rapidly.

Some studies have shown that long sitting time increases the risk of heart disease, cancer, metabolic syndrome, and diabetes [12,13,14], in addition, a reduction in sitting time leads to increased HDL cholesterol levels [15], reduced pain, and improved mental health [16]. Others showed the opposite results [17,18]. However, the “causal” relationship remains unclear. Moreover, recent studies have shown that even at the same total sitting time, there were differences in health risks depending on whether or not the sedentary behaviors were interrupted [6,19]. That is, it has been found that those who rarely interrupt continuous sitting behavior have a negative impact on biomarkers of cardiovascular metabolic disease compared to those who frequently interrupt sitting behavior [20].

Therefore, this study uses the “Granger causality test” [21] to predict the long-term correlation of physical and mental factors in the elderly. Here, the Granger causality test is used to predict causality by controlling the preliminary values of each variable and examining the mutual delay effects between them. If the X→Y and Y→X cross-lag effects are both significant, a two-way causal relationship is estimated. In addition, if only one cross delay effect is important, a bi-directional causal relationship is predicted, and if both cross delay effects are not significant, it is assumed that there is no causal relationship between the variables.

This study aims to clarify the long-term relationship between PD and the rate of CSB. To define our purpose, we assumed the following hypothesis: PD and CSB may interact with each other.

## 2. Materials and Methods

### 2.1. Study Design

In this second analysis from our previous reports [22,23], we conducted a longitudinal study using “three-wave” panel data. We adopted the following as proxy variables: CSB (continuous sedentary behaviors, based on the ratio of 1.5 METs sessions or more continuing for over 30 min) and PD (psychological distress) that are based on the rate of 1.5 METs sessions or more that continued for over 30 min and the Kessler 6 scale (K6) scores, respectively. To define our purpose, we adopted the following hypothesis: In the short term, improvement of PD causes a decreased rate of CSB, but in the long term it is the reverse. The survey methods were assessed using a tri-axial accelerometer and self-administered questionnaire.

This study was approved by the Shikoku Medical College Ethic Screening Committee (approval number: H27-3: May 13, 2016; H27-7: May 12, 2017; H28-6: April 06, 2018), and written informed consent was obtained from each subject.

### 2.2. Participants

The initial survey involved 96 healthy elderly persons who participated at a health club of college A in Utazu, Japan (approximate population of 18,450). As previously described [22], we conducted the study from 20 July to 10 September 2016 in the first phase. Since 3 of 96 people canceled and 7 people did not reach the standard measurements of physical activity, we excluded them from analysis. Thus, the remaining 86 respondents were used as reference databases. The second phase involved a similar follow-up survey and was conducted from 20 July to 15 September 2017. Of these respondents, six could not be surveyed. Therefore, we used data based on 80 participants [23]. The third phase involved a similar follow-up survey and was conducted from 29 April to 31 May 2018. Of these respondents, eight could not be surveyed. Therefore, we used data based on 72 participants (72.6 ± 5.4 years, the K6 scores: 2.5 ± 2.3).

### 2.3. Clinical Parameters and Measurement

Anthropometric and body composition parameters were evaluated as confounders based on the following parameters: Age (years), height (cm), body weight (kg), and body mass index (BMI), kg/m^2^) in 2016, 2017 [13,24], and we added the data obtained in the 2018 survey.

### 2.4. Psychological Distress

Data for the K6 scores are cited in our previous papers. “Psychological distress was assessed using six items of the Japanese edition of the K6 scale. The K6 is a self-written questionnaire developed by Kessler as a screening test for psychological distress that could effectively discriminate—it is valid and reliable. By using a simple questionnaire called K6, we can clarify factors related to psychological distress and provide clues for achieving mental and psychological health. Subjects answered six items on a 5-point Likert scale, and responses for each item were transformed to scores ranging from 0 to 4 points”. The questionnaire consisted of six questions: “Over the last month, about how often did you feel: (1) nervous, (2) hopeless, (3) restless or fidgety, (4) so sad that nothing could cheer you up, (5) that everything was an effort, (6) worthless?”. The subjects were requested to respond by choosing from the following: “all of the time” (4 points), “most of the time” (3 points), “some of the time” (2 points), “a little of the time” (1 point), and “none of the time” (0 point), and the total points were the evaluation level. Thus, the score range was 0–24. [16,17,18], and we added the data obtained in the 2018 survey.

### 2.5. Physical Activity

Data regarding physical activity are cited in our previous papers. “We used a triaxle accelerometer to record data (Active Style Pro HJA-750C, Omron Healthcare corporation) for 7 consecutive days. Subjects were asked to wear these tools at all times except when it was not possible, such as while swimming and bathing, and the standard deviation of the data of 10 s is defined as an average value of acceleration. We adopted seven days including Saturday or Sunday to satisfy wearing 10 h or more per day in this analysis.” [22,23,24], and data obtained from the 2018 survey were also added. CSB was defined as (the time of sedentary behavior continuing for more than 30 min per day)/(awake time in one day in minutes) × 100.

### 2.6. Statistical Analyses

We conducted structural equation modeling (SEM) to clarify the causal relationship between PD and CSB. First, to determine whether the variables from the initial survey could sufficiently predict variables from the second and the third survey, PD and CSB in 2016 were compared with those in 2017 and in 2018. Second, to assess the causal relationship between PD and CSB, we adopted the cross-lagged effects models. Third, PD and CSB in 2017 were compared with those in 2018. Fourth, PD and CSB in 2016 were compared with those in 2018. Finally, to measure the fitness of these models, we used χ^2^ (if *p* > 0.05, it was regarded as conforming to the data), adjusted goodness of fit index (AGFI, from 0 to 1, it corresponds to the adjusted determination coefficient in regression analysis, preferably 0.95 or more), comparative fit index (CFI, from 0 to 1, it is an indicator that corrects the influence of the number of data, preferably 0.95 or more), and root mean square error of approximation (RMSEA; an index that expresses the deviation between the distribution of the model and the true distribution as an amount per one degree of freedom, preferably less than 0.05) [25]. The appropriate sample size for the cross-lagged and synchronous effects models has not yet been established. All calculations were performed using SPSS version 25 and AMOS version 25 (IBM).

## 3. Results

Data are calculated and expressed as mean ± standard deviated (SD) values in Table 1. First, as shown in Figure 1, the K6 scores (PD) and CSB in 2016 were highly correlated in 2017 (PD *β* = 0.914, CSB *β* = 0.826; respectively *p* < 0.001) and the K6 scores (PD) and CSB in 2017 were highly correlated in 2018 (PD *β* = 0.845, CSB *β* = 0.829; respectively *p* < 0.001).

Second, as shown in Figure 1, in the cross-lagged effects model, the path of the model from 2016 K6 scores to 2017 CSB was significant (standardization factor; SF: 0.141, *p* < 0.05); however, the reverse was not (SF: 0.001, *p* = 0.990). Therefore, there is a possibility that the K6 scores in 2016 exerted a causal effect (0.141) on CSB in 2017.

Third, as shown in Figure 1, in the cross-lagged effects model, the path of the model from 2017 K6 scores to 2018 CSB was significant (standardization factor; SF: 0.187, *p* < 0.05); however, the reverse was not (SF: 0.055, *p* = 0.506). Therefore, there is a possibility that the K6 scores in 2017 exerted a causal effect (0.187) on CSB in 2018.

Fourth, as shown in Figure 1, in the cross-lagged effects model, the path of the model from 2016 K6 scores to 2018 CSB was not significant (standardization factor; SF: 0.006, *p* = 0.927); however, the reverse was significant (SF: 0.157, *p* < 0.05). Therefore, there is a possibility that CSB in 2016 exerted a causal effect (0.157) on the K6 scores in 2018. In Figure 2, based on the synchronous effects model, in 2018, the path from the K6 scores to CSB was not statistically significant (SF: 0.043, *p* = 0.603) while the reverse was significant (SF: 0.149, *p* = 0.020). Therefore, there is a possibility that the CSB in 2016 exerted a causal effect (0.149) on the K6 scores in 2018.

Finally, to measure the fitness of these models (excluding insignificant paths), the following fitness indexes were obtained, and both had high degrees of fitness: χ^2^ = 2.002 (*p* = 0.849), AGFI = 0.961, CFI = 1.000 and RMSEA = 0.000.

## 4. Discussion

Our study had five major findings (no gender difference for all). First, PD in 2016 was significantly correlated with PD in 2017 and 2018. Similarly, CSB between in 2016, in 2017, and in 2018 were also significantly correlated. These results might suggest that aging is not a significant factor in this study. A precise analysis must use latent change model (LCM) or latent curve model (LCM) [26], which was not performed in this study. This is because when using LCM, it is desirable that the measurement time points are aligned for each individual, but in this study there was a slight shift in the measurement time points for each individual.

Second and third, analysis with the cross-lagged effects model showed that the paths from the K6 in 2016 to CSB in 2017 and from the K6 in 2017 to CSB in 2018 were still significant even if it was influenced by CSB in 2016 and in 2017. However, the paths from CSB in 2016 to the K6 in 2017 and from CSB in 2017 to the K6 in 2018 were not significant. These results suggest that the K6 in 2016 and in 2017 may have had an impact on CSB in 2017 and in 2018 [27]. Thus, improvement in PD may decrease the ratio of CSB one year later when the cross-lagged models are used. This result was consistent with previous study that PD affected CSB one year later [24].

Fourth, analysis with the cross-lagged effects model showed that the path from K6 in 2016 to CSB in 2018 was not significant even if it was influenced by CSB in 2016. However, the path from CSB in 2016 to K6 in 2018 was significant. Analysis with the synchronous effects model showed that the path from K6 in 2016 to CSB in 2018 was not significant even if it was influenced by CSB in 2016. However, the path from CSB in 2016 to K6 in 2018 was significant. These results suggest that CSB in 2016 may have had an impact on K6 in 2018 [27]. That is, it was found that K6 affected the CSB in the short term (one year) in previous study [24], but CSB affected K6 in the long term (2 years) in this research. Thus, improvement in the ratio CSB may decrease K6 two years later when the cross-lagged and synchronous effects models are used. These results are consistent with other studies that reported that large amounts of physical activity reduced the likelihood of depression [28,29,30].

The elucidation of the physiological and biological mechanisms that cause long-term sitting behavior to cause health risks is still not enough. It has been suggested that prolonged sitting behavior may cause metabolic abnormalities, such as uptake of free fatty acids into muscle cells and suppression of lipoprotein lipase activity necessary for HDL generation [31]. In the sitting position, almost no muscle contraction of the leg is observed. This may make cardiovascular metabolism inactive [32]. Furthermore, the mechanism of the relationship between CSB and PD is not understood. Hamer [33] assumed a psychological mechanism between SB and PD. For example, SB worsened PD because it led to social isolation. In the future, it is necessary to elucidate the mechanism of the relationship between CSB and PD.

Finally, in the structural equation model, it was necessary that a non-logical value did not initially occur in the measurement model and to ascertain that the degree of fitness of the model was good before interpreting the results.

Our research has several limitations. First of all, there may be a problem with regression to the mean [26]. Thus, “when conducting a longitudinal survey at two points, the high score of the initial survey is higher than the true score of the survey target, and the low score of the survey is lower than the true score of the subject to be surveyed tend. As a result, the high score of the initial survey will fall further by the second survey, but the low score of the initial survey tends to be higher. To alleviate this problem, observations at three or more time points are required. This is because the observed scores randomly fluctuate about the true score, so if the change is measured at more than three time points it will alleviate the problem of regression to the mean” [34]. Nevertheless, it may be necessary to investigate more than four points.

Second, in this study, we did not carefully investigate the intervention of the third variable. Even if a causal relationship from *x* to *y* is indicated, there is a possibility that it may be affected by an unknown third variable, including socioeconomic variables. For example, variables such as physical health and socioeconomic status can be considered. In previous study [22], the K6 scores did not related to age, BMI (kg/m^2^), working hours (h/day), walking time (min/day), sleep time (h/day), spouse (presence or none), exercise limitation (presence or none), pain in limbs (presence or none), smoking habits (smoker or none), and drinking habits (drinker or none). Future research should incorporate the third variable *z*, which may affect the estimation of causality between *x* and *y*, into the model.

Finally, since the number of samples was less than 100, we used maximum likelihood and simple model [35]. However, in the future, in order to use more complicated models, a larger number of samples should be obtained.

## 5. Conclusions

The results suggest that PD may affect the ratio of CSB one year later, and CSB may affect the ratio of PD two years later. In other words, for the elderly, PD affects the sitting behavior in the short term, but conversely, in the long term, the sitting behavior is likely to affect PD. Decreasing sitting behavior may improve PD in the elderly. In the future, we would like to analyze the precise causal relationship between sitting behavior and PD, taking into account other factors that improve the mental health of the elderly.

## Figures and Tables

**Figure 1 medicina-55-00555-f001:**
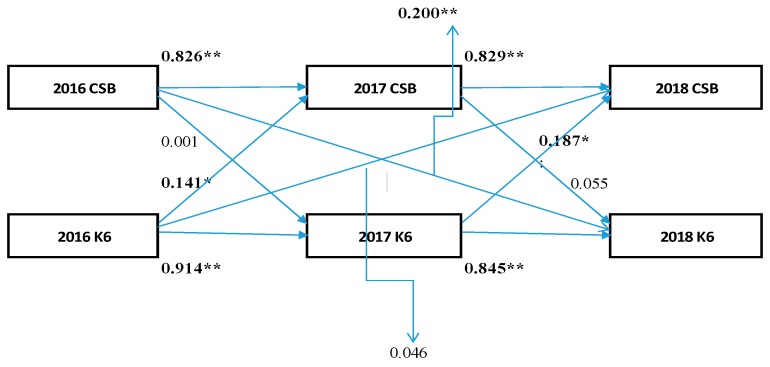
Cross-lagged effects model. ** *p* < 0.001 * *p* < 0.05 CBS: continuous sedentary behaviors, K6:K6 scores.

**Figure 2 medicina-55-00555-f002:**
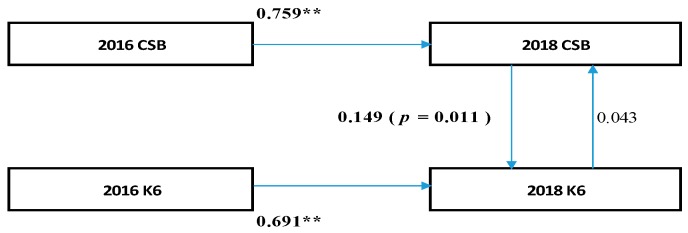
Synchronous effects model. ** *p* < 0.001 CBS: continuous sedentary behaviors, K6:K6 scores.

**Table 1 medicina-55-00555-t001:** Clinical characteristics of enrolled subjects.

	2016					2017					2018				
	Mean	±	SD	Minimum	Maximum	Mean	±	SD	Minimum	Maximum	Mean	±	SD	Minimum	Maximum
Number of subjects	72 (men = 23)														
Age (year)	72.6	±	5.4	65	85										
Height (cm)	157.2	±	9.1	138.3	178.4	157.1	±	9.1	138.1	178.2	157.1	±	9.0	138.0	178.2
Body weight (kg)	55.8	±	10.0	40.3	83.1	56.0	±	9.7	40.5	83.3	56.1	±	10.1	40.2	83.6
BMI (kg/m^2^)	22.9	±	2.4	13.9	29.1	23.1	±	2.6	14.9	29.2	23.6	±	2.6	15.1	29.2
≤1.5 Mets (%/day)	55.1	±	9.7	35.4	79.9	55.5	±	10.9	20.9	75.4	55.7	±	11.0	30.8	86.6
CSB:Interrupted of sedentary behavior (%)	14.7	±	8.5	0.0	40.8	15.8	±	9.5	0.0	45.4	15.1	±	7.9	0.0	39.8
K6 scores	2.5	±	3.1	0	14	2.7	±	3.2	0	14	2.5	±	2.3	0	11

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
