# Peer review of "Long-Term Relationship between Psychological Distress and Continuous Sedentary Behavior in Healthy Older Adults: A Three Panel Study"

_medicina, 2019, doi:10.3390/medicina55090555_

Round 1

Reviewer 1 Report

Thank you very much for giving me the opportunity to revise the article entitled "Long-term Relationship between Psychological Distress and Continuous Sedentary Behavior in Healthy Older Adults: A Three Panel Study". Although the topic is interesting and the statistical analyses are significant strengths of the study, some changes and recommendations should be addressed by authors: 

Introduction:

The introduction section needs to include more information. Fundamentally, authors should include the conceptualization of psychological distress and sedentary behaviour, and how both dimensions have been related to negative health consequences in healthy older adults.

Moreover, authors should include the aims of the study and the related hypotheses.

Materials and Methods

This section should include a subheading of “Participants” in which authors describe the main characteristics of the participants in the study, mainly sociodemographic dimensions and not only clinical variables.

Discussion

Authors should include specific explanatory mechanism for the obtained results. In this sense, authors have only described again the results obtained in the study, but they do not include which type of mechanisms could explain these results.

Further, authors should include a conclusion with much more information, in which the resume the main findings of the study and propose future research lines.

Author Response

Comment 1

Thank you very much for giving me the opportunity to revise the article entitled "Long-term Relationship between Psychological Distress and Continuous Sedentary Behavior in Healthy Older Adults: A Three Panel Study". Although the topic is interesting and the statistical analyses are significant strengths of the study, some changes and recommendations should be addressed by authors:

Introduction:

The introduction section needs to include more information. Fundamentally, authors should include the conceptualization of psychological distress and sedentary behaviour, and how both dimensions have been related to negative health consequences in healthy older adults.

(Answer)

   Thank you for your meaningful feedback. Following the comments, we added the following.

P1, L29-30.

Recent research has shown that oversitting is a health risk factor, even with moderate physical activity [4].

P1, L36-38

Some studies have shown that long sitting time increases the risk of heart disease, cancer, metabolic syndrome and diabetes [6-8], in addition, a reduction in sitting time leads to increase HDL cholesterol levels [9] and improves pain and mental health [10].

P2, L1-3.

That is, it has been found that those who rarely interrupt continuous sitting behavior have a negative impact on biomarkers of cardiovascular metabolic disease compared to those who frequently interrupt sitting behavior [15].

Moreover, authors should include the aims of the study and the related hypotheses.

(Answer)

   Thank you for your meaningful feedback. Following the comments, we added the following.

P2, L59-60.

This study aims to clarify the long-term relationship between PD and the rate of CSB. To define our purpose, we assumed the following hypothesis: PD and CSB may interact with each other.

Materials and Methods

This section should include a subheading of “Participants” in which authors describe the main characteristics of the participants in the study, mainly sociodemographic dimensions and not only clinical variables.

(Answer)

   Thank you for your meaningful feedback. The manuscript has been revised as follows in accordance with your comments.

P2, L73-83.

2.2. Participants

The initial survey involved 96 healthy elderly persons who participated at a health club of college A in Utazu, Japan (approximate population of 18,450). As previously described [17], we conducted the study from July 20 to September 10, 2016 in the first phase. Since 3 of 96 people canceled and 7 people did not reach the standard measurements of physical activity, we excluded them from analysis. Thus, the remaining 86 respondents were used as reference databases. The second phase involved a similar follow-up survey and was conducted from July 20 to September 15, 2017. Of these respondents, six could not be surveyed. Therefore, we used data based on 80 participants [18]. The third phase involved a similar follow-up survey and was conducted from April 29 to May 31, 2018. Of these respondents, eight could not be surveyed. Therefore, we used data based on 72 participants (72.6 ± 5.4 years, the K6 scores: 2.5 ± 2.3).

Discussion

Authors should include specific explanatory mechanism for the obtained results. In this sense, authors have only described again the results obtained in the study, but they do not include which type of mechanisms could explain these results.

(Answer)

   Thank you for your meaningful feedback. Following the comments, we added the following.

P5, L177-182.

The elucidation of the physiological and biological mechanisms that cause long-term sitting behavior causes health risks is still not enough. It has been suggested that prolonged sitting behavior may cause metabolic abnormalities such as uptake of free fatty acids into muscle cells and suppression of lipoprotein lipase activity necessary for HDL generation [23]. In sitting position, almost no muscle contraction of the leg is observed. This may make cardiovascular metabolism inactive [24].

Further, authors should include a conclusion with much more information, in which the resume the main findings of the study and propose future research lines.

(Answer)

   Thank you for your meaningful feedback. Following the comments, we added the following.

P6, L204-208.

In other words, for the elderly, mental health affects the sitting behavior in the short-term, but conversely, in the long-term, the sitting behavior is likely to affect the mental health. Decreasing sitting behavior may improve the mental health in the elderly. In the future, I would like to analyze the precise causal relationship between sitting behavior and mental health, taking into account other factors that improve the mental health of the elderly.

Reviewer 2 Report

In the paper, the authors examined the relation between psychological health and continuous sedentary behavior in a longitudinal data set. A mutual relation between the two variables has been found, which has theoretical and practical implications. However, there are some weaknesses in the analyses and problems in data presentation.

1.In the second paragraph(p.1), the literature on health outcomes of sedentary behavior should be specified, e.g., what health risks (physical or mental health outcomes)were increased by the reduction in sitting time?

2. The definition of causal relationship may be problematic. Granger Causal Relationship may not be suitable for medical research. Although this approach has values in predicting health outcomes, Granger causality test does not reveal a true causal relationship or indicate underlying mechanisms between two variables (because that there were no manipulations in the study). The methodology in the present study was observational in nature. Therefore, the current findings are better to be described as the correlations that have predicting values.

3. Related to the last point, a bi-directional relationship may indicate a underlying mechanism, or some factors influencing both variables, e.g., physical health and socioeconomic status.

4. The authors did not explicitly state the reasons for the hypotheses. Given the literature is equivocal, it was unclear why a positive relation between CSB and PD was hypothesized.

5. Please spelled out the abbreviations and explained the terms when they appeared in text for the first time, e.g., CSB, PD, and K6

6. What are psychometric characteristics of the K6 scale?

7. Were there sex differences in CSB, PD, and their relation?

8. In Discussion, correlations between PD in 2016, 2017, and 2018 can not be used to demonstrate that aging was not a contributing factor. A developmental trajectory should be delineate to investigate the effect of age. The same problem is for CSB.

Author Response

Comment 2

In the paper, the authors examined the relation between psychological health and continuous sedentary behavior in a longitudinal data set. A mutual relation between the two variables has been found, which has theoretical and practical implications. However, there are some weaknesses in the analyses and problems in data presentation.

1.In the second paragraph(p.1), the literature on health outcomes of sedentary behavior should be specified, e.g., what health risks (physical or mental health outcomes)were increased by the reduction in sitting time?

(Answer)

   Thank you for your meaningful feedback. The manuscript has been revised as follows in accordance with your comments.

P1, L36-38.

Some studies have shown that long sitting time increases the risk of heart disease, cancer, metabolic syndrome and diabetes [6-8], in addition, a reduction in sitting time leads to increase HDL cholesterol levels [9] and improves pain and mental health [10]. Others showed the opposite results [11, 12].

The definition of causal relationship may be problematic. Granger Causal Relationship may not be suitable for medical research. Although this approach has values in predicting health outcomes, Granger causality test does not reveal a true causal relationship or indicate underlying mechanisms between two variables (because that there were no manipulations in the study). The methodology in the present study was observational in nature. Therefore, the current findings are better to be described as the correlations that have predicting values.

(Answer)

   Thank you for your meaningful feedback. The manuscript has been revised as follows in accordance with your comments.

P2, L52-60.

Therefore, this study uses the “Granger Causality Test” [9] to predict the long-term correlation of physical and mental factors in the elderly. Here, the Granger Causality Test is used to predict causality by controlling the preliminary values of each variable and examining the mutual delay effects between them. If the X → Y and Y → X cross-lag effects are both significant, a two-way causal relationship is estimated. In addition, if only one cross delay effect is important, a bi-directional causal relationship is predicted, and if both cross delay effects are not significant, it is assumed that there is no causal relationship between the variables.

Related to the last point, a bi-directional relationship may indicate a underlying mechanism, or some factors influencing both variables, e.g., physical health and socioeconomic status.

(Answer)

   Thank you for your meaningful feedback. Following the comments, we added the following.

P5, L197.

an unknown third variable including socioeconomic variables.

The authors did not explicitly state the reasons for the hypotheses. Given the literature is equivocal, it was unclear why a positive relation between CSB and PD was hypothesized.

(Answer)

   Thank you for your meaningful feedback. Following the comments, we added the following.

P1, 36-40, L2, L41-43.

Some studies have shown that long sitting time increases the risk of heart disease, cancer, metabolic syndrome and diabetes [6-8], in addition, a reduction in sitting time leads to increase HDL cholesterol levels [9] and improves pain and mental health [10]. Others showed the opposite results [11, 12]. However, the causal relationship remains unclear. Moreover, recent studies have showed that even at the same total sitting time, there were differences in health risks depending on whether or not the sedentary behaviors were interrupted [13, 14]. That is, it has been found that those who rarely interrupt continuous sitting behavior have a negative impact on biomarkers of cardiovascular metabolic disease compared to those who frequently interrupt sitting behavior [15].

P2, L59-60.

This study aims to clarify the long-term relationship between PD and the rate of CSB. To define our purpose, we assumed the following hypothesis: PD and CSB may interact with each other.

Please spelled out the abbreviations and explained the terms when they appeared in text for the first time, e.g., CSB, PD, and K6

(Answer)

   Thank you for your meaningful feedback. Following the comments, we added the following.

P3, L118-124.

adjusted goodness of fit index (AGFI: from 0 to 1, it corresponds to the adjusted determination coefficient in regression analysis, preferably 0.95 or more), comparative fit index (CFI: from 0 to 1, it is an indicator that corrects the influence of the number of data, preferably 0.95 or more), and root mean square error of approximation (RMSEA: it is an index that expresses the deviation between the distribution of the model and the true distribution as an amount per one degree of freedom, preferably less than 0.05)

P2, L64-66.

continuous sedentary behaviors: based on the ratio of 1.5 METs sessions or more continuing for over 30 minutes (CSB) and Psychological distress (PD)

P2, L67.

the Kessler 6 scale (K6)

What are psychometric characteristics of the K6 scale?

(Answer)

   Thank you for your meaningful feedback. Following the comments, we added the following.

P3, L92-94.

By using a simple questionnaire called K6, we can clarify factors related to psychological distress and provide clues for achieving mental and psychological health.

Were there sex differences in CSB, PD, and their relation?

(Answer)

   Thank you for your meaningful feedback. Following the comments, we added the following. There is no gender difference for all.

P5, L157.

(no gender difference for all)

In Discussion, correlations between PD in 2016, 2017, and 2018 can not be used to demonstrate that aging was not a contributing factor. A developmental trajectory should be delineate to investigate the effect of age. The same problem is for CSB.

(Answer)

   Thank you for your meaningful feedback. Following the comments, we added the following.

P5, L159-161.

These results might suggest that aging is not a significant factor in this study. A precise analysis must use Latent Change Model or Latent Curve Model [21] which was not performed in this study.

Round 2

Reviewer 1 Report

Although authors have made an effort improving the manuscript, few issues should remains unclear. 

The introduction section remains very short. The information included by authors has been very scarce and is not a sufficient background for possible readers of the manuscript. More information in this section is needed, together with more much elaboration from previous literature in the topic. 

The lack of more information regarding sociodemographic and clinical characteristics of participants is a serious weakness of the manuscript, taking into account that they could act as confounders of the obtained results. This fact should be included and justified as a serious limitation of the study. 

The mechanisms included in the discussion section seem to be not related to psychological distress. Authors should identify the underlying mechanisms in the association between sedentary behavior and psychological distress. If it is not possible, it is necessary that authors propose some mechanisms based n previous literature in the topic of study. 

Author Response

Comment 1.

Although authors have made an effort improving the manuscript, few issues should remains unclear.

The introduction section remains very short. The information included by authors has been very scarce and is not a sufficient background for possible readers of the manuscript. More information in this section is needed, together with more much elaboration from previous literature in the topic.

(Answer)

   Thank you for your meaningful feedback. Following the comments, we added the following.

P.1, L29-34.

Sedentary behavior has become incorporated into many aspects of both workplace and family life in modern societies. In particularly, the sedentary behavior of Japanese is the highest in the world [1, 2]. Moreover, sedentary behavior increases with age [3], and is a risk factor for various adverse health outcomes [4, 5]. In our study, sedentary behavior is defined not as “physical inactivity” (the definition of the World Health Organization [6]), but as “any waking behavior characterized by an energy expenditure ≤1.5 metabolic equivalents (METs) while in a sitting or reclining posture” [7].

The lack of more information regarding sociodemographic and clinical characteristics of participants is a serious weakness of the manuscript, taking into account that they could act as confounders of the obtained results. This fact should be included and justified as a serious limitation of the study.

(Answer)

   Thank you for your meaningful feedback. Following the comments, we added the following.

P.5, L199-203.

For example, variables such as physical health and socioeconomic status can be considered. In previous study [24], the K6 scores did not related to age, BMI (kg/m2), working hours (hours/day), walking time (minutes/day), sleep time (h/day), spouse (presence or none), exercise limitation (presence or none), pain in limbs (presence or none), smoking habits (smoker or none), and drinking habits (drinker or none).

The mechanisms included in the discussion section seem to be not related to psychological distress. Authors should identify the underlying mechanisms in the association between sedentary behavior and psychological distress. If it is not possible, it is necessary that authors propose some mechanisms based n previous literature in the topic of study.

(Answer)

   Thank you for your meaningful feedback. Following the comments, we added the following.

P.5, L.181-184.

Furthermore, the mechanism the relationship between CSB and PD is not understood. Hamer [35] assumed psychological mechanism between SB and PD. For example, SB worsened PD because it led to social isolation. In the future, it is necessary to elucidate the mechanism of the relationship between CSB and PD.

Reviewer 2 Report

The authors have addressed the concerns that were raised in the comments. Some points in Introduction have been clarified, and potential confounding effects were tested and discussed. Overall, the revised manuscript have been improved, which is recommended for publication.

There is just one minor suggestion for the authors to consider before finalizing the paper:

In the first paragraph of Discussion (p.5), the authors stated that LCM should be used, but was not performed in the study. It would be better to explain why this analysis was not performed, which could be a future direction.

Author Response

Comment 2.

The authors have addressed the concerns that were raised in the comments. Some points in Introduction have been clarified, and potential confounding effects were tested and discussed. Overall, the revised manuscript have been improved, which is recommended for publication.

There is just one minor suggestion for the authors to consider before finalizing the paper:

In the first paragraph of Discussion (p.5), the authors stated that LCM should be used, but was not performed in the study. It would be better to explain why this analysis was not performed, which could be a future direction.

(Answer)

   Thank you for your meaningful feedback. Following the comments, we added the following.

P.5.L.158-160.

This is because, when using LCM, it is desirable that the measurement time points are aligned for each individual, but in this study there was a slight shift in the measurement time points for each individual.
